# GLP-1 Receptor Agonists in the Treatment of Patients with Type 2 Diabetes and Chronic Kidney Disease

**Joshua J. Neumiller** [1,2,*], **Radica Z. Alicic** [2,3] **and Katherine R. Tuttle** [2,3,4]

1. Department of Pharmacotherapy, College of Pharmacy and Pharmaceutical Sciences, Washington State University, Spokane, WA 99210, USA
2. Providence Medical Research Center, Providence Health Care, Spokane, WA 99204, USA; radica.alicic@providence.org (R.Z.A.); katherine.tuttle@providence.org (K.R.T.)
3. Department of Medicine, University of Washington School of Medicine, Seattle, WA 98195, USA
4. Nephrology Division, Kidney Research Institute and Institute of Translational Health Sciences, University of Washington, Seattle, WA 98104, USA
* Correspondence: jneumiller@wsu.edu; Tel.: +1-509-368-6756

**Abstract:** Diabetic kidney disease (DKD) represents an important diabetes (DM) complication associated with significant impacts on morbidity, mortality, and quality of life. Recent evidence from cardiovascular and kidney outcome trials has dramatically impacted the standard of care for patients with DKD. While agents from the glucagon-like peptide-1 (GLP-1) receptor agonist class are known for their atherosclerotic cardiovascular disease (ASCVD) benefits, growing mechanistic and clinical evidence supports the benefit of GLP-1 receptor agonist therapy on progression of DKD. GLP-1 receptor activation is associated with anti-inflammatory and antifibrotic effects in the kidney, providing a plausible mechanism for kidney protection. Based on currently available clinical trial evidence, guidelines recommend the use of GLP-1 receptor agonists to mitigate ASCVD risk in patients with type 2 diabetes (T2D). Furthermore, based on secondary outcome data for kidney disease, GLP-1 receptor agonists are recommended as an option to mitigate kidney and ASCVD risk in patients with T2D and DKD who require intensification of glycemic control or for those who cannot take a sodium-glucose cotransporter-2 (SGLT2) inhibitor due to side effects or advanced stage DKD. Ongoing dedicated kidney disease outcome trials will further inform the role of GLP-1 receptor agonists in DKD management. This review discusses current considerations for GLP-1 receptor agonist use in patients with T2D and DKD.

**Keywords:** albuminuria; atherosclerotic cardiovascular disease; dulaglutide; exenatide; lixisenatide; liraglutide; semaglutide

## 1. Introduction

The number of people living with diabetes (DM) continues to increase steadily worldwide [1]. An estimated 537 million adults had DM in 2021, equating to approximately 10.5% of all adults worldwide [1]. The number of people living worldwide with DM is projected to increase to 783 million cases by the year 2045 [1]. Incredibly, while the world population is projected to increase by 20% by 2045, the number of people living with DM is anticipated to increase by 46% [1]. Of the total number of people living with DM, the large majority (~95%) have type 2 diabetes (T2D) [2]. A major goal of T2D management is optimization of glycemic control to prevent or delay vascular complications that markedly increase risk for morbidity and mortality [3,4]. Chronic kidney disease (CKD) in DM, also known as diabetic kidney disease (DKD), is a particularly challenging complication given the impact of DKD on health outcomes, quality of life, and day-to-day medication management [5]. In terms of health-related outcomes, DKD increases the risk of all-cause and cardiovascular (CV) death by five- to six-fold [6,7]. Additionally, when compared to people without DM, patients with DKD have a greater likelihood of developing coronary

heart disease and heart failure (HF) [5]. Fortunately, findings from recent large cardiovascular outcome trials (CVOTs) and dedicated kidney outcome trials with agents from the sodium-glucose cotransporter-2 (SGLT2) inhibitor, non-steroidal mineralocorticoid receptor antagonist (MRA), and glucagon-like peptide-1 (GLP-1) receptor agonist classes have identified important CV and kidney benefits that have changed the standard of care for treating people with DKD [5,8].

This review will focus on current evidence for the use of GLP-1 receptor agonists in patients with T2D and DKD to improve CV and kidney outcomes. Additionally, it provides a brief discussion of proposed mechanisms of kidney protection with GLP-1 receptor agonists and current evidence-based recommendations for their clinical use to improve glycemic, CV, and kidney outcomes in patients with T2D.

## 2. Proposed Mechanisms for Kidney Benefit with GLP-1 Receptor Agonists

GLP-1 receptor agonists were initially developed as glucose-lowering therapies, which have demonstrated significant efficacy for glycemia lowering with low risk of hypoglycemia [8,9]. Although glucose lowering per se may mitigate kidney injury from diabetes, the beneficial effects on atherosclerotic cardiovascular disease (ASCVD) and DKD have mostly been attributed to non-glycemic actions. [10–17]. GLP-1 receptor agonist therapy also results in improvements for other shared CV and DKD risk factors such as reductions in body weight and blood pressure [8]. A systematic review and meta-analysis reported a mean weight loss with GLP-1 receptor agonist therapy of approximately three kilograms [18]. Newer agents within the class, however, are associated with the potential for considerably greater weight loss [19–21], with injectable semaglutide recently joining liraglutide as GLP-1 receptor agonists that carry an obesity indication [22,23]. GLP-1 receptor agonist treatment was additionally associated with mean systolic blood pressure reductions in the range of 3–4 mmHg in large CVOTs [24]. Figure 1 provides a summary of putative mechanisms of GLP-1 receptor agonist benefit on DKD and ASCVD outcomes [14,25–29].

The pathogenesis of DKD involves damage to the kidney via multiple metabolic and hemodynamic mechanisms, including inflammation and fibrosis, that lead to a progressive decline in estimated glomerular filtration rate (eGFR) [28]. The production of advanced glycation end-products in conjunction with increases in oxidative stress, insulin resistance and resultant hyperglycemia promotes immune system activation early in the course of DM [15,25,26]. Immune system activation subsequently exacerbates inflammation within the kidney over time and possibly activates resident kidney T cell populations [30]. Inflammatory cell invasion subsequently promotes growth factors and pro-fibrotic cytokine upregulation, contributing to fibrosis and damage to the kidney [26,31,32].

GLP-1 receptor agonists have demonstrated anti-inflammatory, antioxidant, and antifibrotic effects, which may explain their beneficial effects on the kidney in T2D [28]. Treatment with native GLP-1 and exenatide reduced levels of multiple markers of inflammation and oxidative stress in adults with T2D (e.g., interleukin-6, interleukin-1β, monocyte chemoattractant protein-1, prostaglandins, serum amyloid A, tumor necrosis factor-α, Toll-like receptors and circulating mononuclear cells) [33]. Furthermore, GLP-1 receptor agonist treatment has demonstrated anti-inflammatory and antioxidative effects in experimental models of DKD, leading to reductions in proteinuria and indicators of endothelial cell injury [34,35]. The anti-inflammatory and antifibrotic effects of GLP-1 receptor agonists in the kidney are believed to be largely mediated by GLP-1 receptor activation (Figure 2) [28]. Current studies demonstrate a variable expression of GLP-1 receptors in different structures of kidney including within the arterial vasculature, glomerular capillaries, endothelial cells, macrophages, juxtaglomerular cells and possibly proximal tubules [36–39].

It has been hypothesized that GLP-1 receptor agonists may also convey benefits in the setting of DKD through promotion of natriuresis and urine alkalization (Figure 1) [40]. Acute infusion of GLP-1 agonists in rodents induces natriuresis and diuresis [41,42]. Some human studies demonstrate similar effect on natriuresis and diuresis [43,44]. For example, studies in overweight adults with and without T2D have reported increased natriuresis

resulting from infusion of exenatide when compared to placebo [43,45]. These effects do not appear to be sustained long term, however, with a 12-week liraglutide treatment trial not finding sustained changes in sodium and fluid balance, with kidney sodium excretion matching sodium intake at the end of the 12-week trial period [46]. A proposed mechanism for natriuresis and diuresis is via the inhibition of the sodium hydrogen exchanger (NHE3) [47].

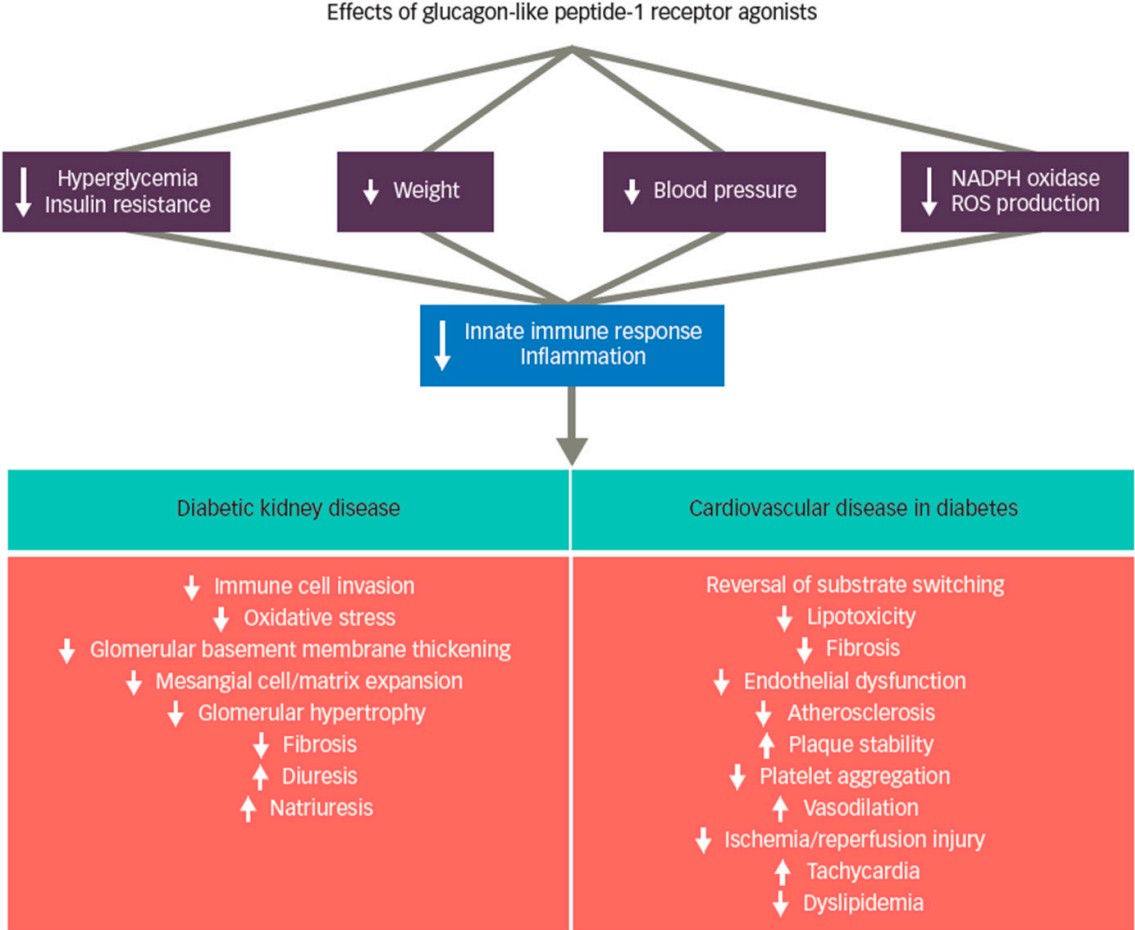

**Figure 1.** Putative mechanisms of GLP-1 receptor agonist therapies on DKD and ACVD. **Legend:** Systemic effects of glucagon-like peptide-1 (GLP-1) receptor agonist treatment include reduction in hyperglycemia, insulin resistance, body weight, blood pressure, reactive oxygen species (ROS) production, and nicotinamide adenine dinucleotide phosphate oxidase (NADPH) activity, resulting in modulation of the inflammatory response. Proposed effects in the kidney are principally related to suppression of inflammation and specifically include suppression of oxidative stress, reduced fibrosis, and blockade of immune cell infiltration. In the heart, GLP-1 receptor agonist therapy also reduces inflammation, and appears to benefit both endothelial dysfunction and dyslipidemia. In the carotid body, GLP-1 signaling is associated with suppressed sympathetic and arterial blood pressure responses, offering a plausible hypothesis for blood pressure reduction. Tachycardia may be mediated by direct agonism of GLP-1 receptors expressed by cells within the autonomic nervous system. Reprinted with permission from Ref. [29].

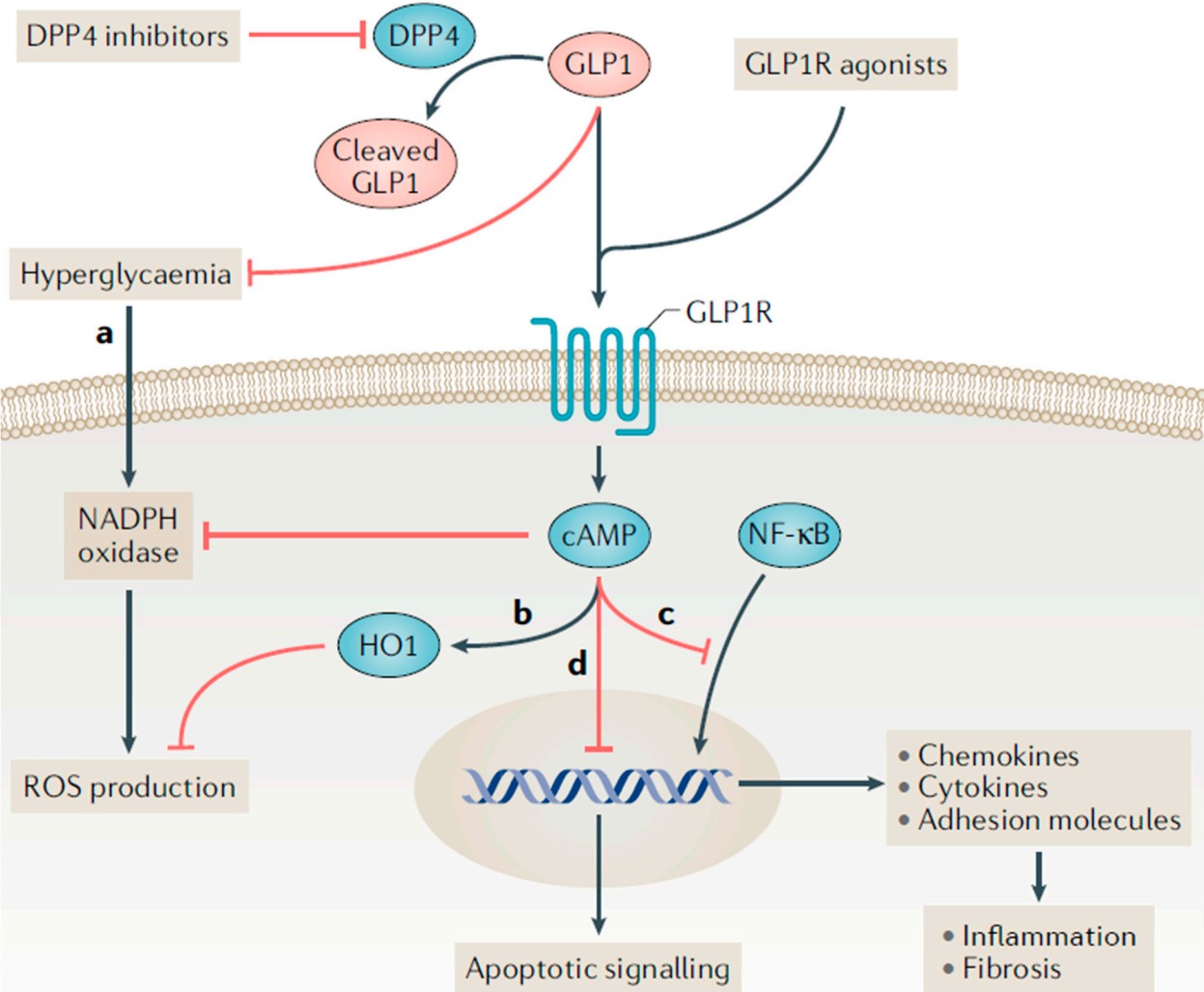

**Figure 2.** Proposed incretin signaling pathways in kidney cells. **Legend:** The glucagon-like peptide 1 receptor (GLP-1R) is a G protein-coupled receptor that can be activated by endogenously produced GLP-1 and synthetic GLP-1 receptor agonists. Dipeptidyl peptidase 4 (DPP4) inhibitors indirectly facilitate GLP-1R activation by preventing the rapid degradation of endogenous GLP-1 in circulation. (**a**,**b**) GLP-1 receptor agonists may reduce the production of reactive oxygen species (ROS) through receptor-mediated and non-receptor-mediated mechanisms. Haem oxygenase 1 (HO1) is upregulated by GLP-1 receptor agonists and protects against oxidative stress in ischemia-reperfusion injury. (**c**) GLP-1R activation inhibits the binding of nuclear factor-κB (NF-κB) p65 to its target genes, which may reduce the downstream expression of chemokines, cytokines (such as tumor necrosis factor, IL-1β, IL-6 and transforming growth factor-β), and pro-fibrotic factors and adhesion molecules (such as intercellular adhesion molecule 1, vascular cell adhesion molecule 1, and E-selectin). (**d**) Administering a DPP-4 inhibitor to mice reduced the ratio of the apoptosis regulator BAX to the apoptosis regulator BCL-2, and the ratio of BCL-2-like protein 11 to BCL-2; this effect suggests a decrease in apoptosis. Whether this anti-apoptotic effect is directly medicated by DPP4 inhibitors or by GLP-1R activation is uncertain. Reprinted with permission from Ref. [28].

## 3. Kidney Disease Outcomes with GLP-1 Receptor Agonists

In 2008, the United States (U.S.) Food and Drug Administration (FDA) published a Guidance for Industry requiring that newly approved glucose-lowering agents undergo cardiovascular safety testing in dedicated CVOTs [48]. While initially intended to establish safety, large CVOTs have fortuitously identified agents within the SGLT2 inhibitor and GLP-1 receptor agonist classes with CV and/or kidney protective effects. Indeed, CVOTs for

several GLP-1 receptor agonists reported significant benefits on primary MACE outcomes and, for some agents, on prespecified secondary kidney disease outcomes (Table 1) [49–54]. In addition to the six GLP-1 receptor agonist CVOTs summarized in Table 1, a clinical trial in patients with T2D and moderate-to-severe DKD was also conducted with the agent dulaglutide [55]. Altogether, currently available kidney outcome evidence is limited to secondary outcomes, with primary kidney outcome data pending from an ongoing kidney outcome trial with injectable semaglutide (NCT03819153) [56]. For the discussion below, DKD is used interchangeably to describe the presence of CKD in patients with T2D. Additionally, while additional small studies looking at kidney outcomes with GLP-1 receptor agonists are reported in the literature, the following discussion focuses primarily on findings from large prospective outcome trials.

**Table 1.** GLP-1 Receptor Agonist Cardiovascular outcome trials (CVOTs) [49–54].

| | REWIND (N = 9901) | EXSCEL (N = 14,752) | ELIXA (N = 6068) | LEADER (N = 9340) | SUSTAIN-6 (N = 3297) | PIONEER-6 (N = 3183) |
|---|---|---|---|---|---|---|
| **Agent** | Dulaglutide | Exenatide XR | Lixisenatide | Liraglutide | Semaglutide | Oral semaglutide |
| **Median follow-up (years)** | 5.4 | 3.2 | 2.1 | 3.8 | 2.1 | 1.3 |
| **Metformin use (%)** | 81 | 77 | 66 | 76 | 73 | 77 |
| **Prior CVD (%)** | 32 | 73 | 100 | 81 | 60 | 85 |
| **Mean baseline A1C (%)** | 7.4 | 8.0 | 7.7 | 8.7 | 8.7 | 8.2 |
| **Primary CV Outcome *** | 3-point MACE 0.88 (0.79–0.99) | 3-point MACE 0.91 (0.83–1.00) | 4-point MACE 1.02 (0.89–1.17) | 3-point MACE 0.87 (0.78–0.97) | 3-point MACE 0.74 (0.58–0.95) | 3-point MACE 0.79 (0.57–1.11) |
| *Key Secondary Outcomes **** | | | | | | |
| **CV death** | 0.91 (0.78–1.06) | 0.88 (0.76–1.02) | 0.98 (0.78–1.22) | 0.78 (0.66–0.93) | 0.98 (0.65–1.48) | 0.49 (0.27–0.92) |
| **MI** | 0.96 (0.79–1.15) | 0.97 (0.85–1.10) | 1.03 (0.87–1.22) | 0.86 (0.73–1.00) | 0.74 (0.51–1.08) | 1.18 (0.73–1.90) |
| **Stroke** | 0.76 (0.61–0.95) | 0.85 (0.70–1.03) | 1.12 (0.79–1.58) | 0.86 (0.71–1.06) | 0.61 (0.38–0.99) | 0.74 (0.35–1.57) |
| **HF hospitalization** | 0.93 (0.77–1.12) | 0.94 (0.78–1.13) | 0.96 (0.75–1.23) | 0.87 (0.73–1.05) | 1.11 (0.77–1.61) | 0.86 (0.48–1.55) |
| **All-cause mortality** | 0.90 (0.80–1.01) | 0.86 (0.77–0.97) | 0.94 (0.78–1.13) | 0.85 (0.74–0.97) | 1.05 (0.74–1.50) | 0.51 (0.31–0.84) |
| **Worsening nephropathy** | 0.85 (0.77–0.93) | - | - | 0.78 (0.67–0.92) | 0.64 (0.46–0.88) | - |

\* Outcome data represented as hazard ratio (HR) and 95% confidence interval (95% CI). **Abbreviations:** A1C, glycated hemoglobin; CV, cardiovascular; CVD, cardiovascular disease; GLP-1 RAs, glucagon-like peptide-1 receptor agonists; HF, heart failure; MACE, major adverse cardiovascular events; MI, myocardial infarction; XR, extended release.

### 3.1. Dulaglutide

The Researching Cardiovascular Events with a Weekly Incretin in Diabetes (REWIND) CVOT enrolled 9901 participants with T2D [49]. REWIND participants were followed for a median of 5.4 years (Table 1). In contrast to earlier CVOTs completed with agents from the GLP-1 receptor agonist class, less than one-third of participants in REWIND had a history of CV disease, with approximately two-thirds of the study cohort constituting a "primary prevention" population with established CV risk factors. The primary three-point MACE composite outcome (inclusive of first occurrence of non-fatal myocardial infarction, non-fatal stroke, or death from CV causes) was significantly reduced with dulaglutide treatment when compared to placebo (hazard ratio (HR): 0.88; 95% confidence interval (CI): 0.79–0.99; *p* = 0.026). Based on findings from REWIND, dulaglutide receved an expanded indication in the U.S. to reduce the risk of MACE in adults with T2D with established CV disease or multiple CV risk factors [57].

REWIND included an exploratory composite kidney outcome that included first occurrence of new severely increased albuminuria (UACR > 300 mg/g), a sustained $\geq$30% decline in eGFR from baseline, or initiation of kidney replacement therapy (KRT) [49]. Over the median follow-up of 5.4 years, the composite kidney outcome occurred in 17.1% versus 19.6% of study participants in the dulaglutide and placebo groups, respectively (HR: 0.85; 95% CI: 0.77–0.93; $p$ = 0.0004; Table 2) [49]. When the components of the composite kidney outcome were examined individually, the clearest benefit was observed for new onset of severely increased albuminuria (HR: 0.77; 95% CI: 0.68–0.87; $p$ < 0.0001), as may be expected for a study population selected for CV risk rather than CKD risk [49].

**Table 2.** Select Kidney Outcomes from GLP-1 Receptor Agonist Clinical Trials [49,53,55,58–61].

| Agent | Secondary Kidney Outcomes |
|---|---|
| Dulaglutide | **REWIND**<br>• Fewer composite kidney outcome events (new onset of severely increased albuminuria (UACR > 300 mg/g), a sustained decline in eGFR of $\geq$30% from baseline, or initiation of KRT) with dulaglutide treatment (HR: 0.85; 95% CI: 0.77–0.93; $p$ = 0.0004)<br><br>**AWARD-7**<br>• eGFR was higher at 52 weeks with dulaglutide 1.5 mg (LSM 34.0 mL/min/1.73 m$^2$ (SE 0.7)) when compared to insulin glargine (LSM 31.3 mL/min/1.73 m$^2$ (SE 0.7)) treatment ($p$ = 0.005)<br>• eGFR was higher at 52 weeks with dulaglutide 0.75 mg (LSM 33.8 mL/min/1.73 m$^2$ (SE 0.7)) when compared to insulin glargine (LSM 31.3 mL/min/1.73 m$^2$ (SE 0.7)) treatment ($p$ = 0.009)<br>• Dulaglutide 1.5 mg group had lower risk for kidney composite outcome ($\geq$40% eGFR decline, progression to ESKD) when compared to participants randomized to insulin glargine over 1 year (HR: 0.45; 95% CI: 0.20–0.97; $p$ = 0.04)<br>   o In the subset of participants with severely increased albuminuria, risk for the composite kidney outcome was significantly improved with dulaglutide 1.5 mg versus insulin glargine over 1 year (HR: 0.25; 95% CI: 0.10–0.68; $p$ = 0.006) |
| Exenatide | **EXSCEL**<br>• New macroalbuminuria occurred in 2.2% of participants in the exenatide group and in 2.5% of participants in the placebo groups (HR: 0.87; 95% CI: 0.70–1.07)<br>• No difference between treatment groups was observed for either of two prespecified kidney composite outcomes |
| Lixisenatide | **ELIXA**<br>• In participants with severely increased albuminuria, the placebo-adjusted least-squares mean percentage change from baseline in UACR was −39.18% (−68.53 to −9.84; $p$ = 0.04) in participants with severely increased albuminuria<br>• Lixisenatide treatment decreased risk of new onset severely increased albuminuria when adjusted for baseline A1C (HR: 0.808; 95% CI: 0.660–0.991; $p$ = 0.0404) or baseline and on-trial A1C (HR: 0.815; 95% CI: 0.665–0.999; $p$ = 0.0491)<br>• No significant differences between treatment groups in eGFR decline were observed |
| Liraglutide | **LEADER**<br>• The composite kidney outcome (new-onset persistent macroalbuminuria, persistent doubling of serum creatinine, ESKD, or kidney-related death) was reduced with liraglutide treatment (HR: 0.78; 95% CI: 0.67–0.92; $p$ = 0.003)<br>• eGFR change over 36 months was −7.44 mL/min/1.73 m$^2$ in the liraglutide group vs. −7.82 mL/min/1.73 m$^2$ in the placebo group<br>• UACR increased less in the liraglutide group over 36 months, yielding a 17% lower UACR compared to placebo (HR: 0.83; 95% CI: 0.79–0.88; $p$ < 0.001) |
| Semaglutide | **SUSTAIN-6**<br>• Occurrence of new or worsening nephropathy was reduced with semaglutide treatment (HR: 0.64; 95% CI: 0.46–0.88; $p$ = 0.005) |

**Abbreviations:** A1C, glycated hemoglobin A1c; CI, confidence interval; eGFR, estimated glomerular filtration rate; HR, hazard ratio; KRT, kidney replacement therapy; LSM, least squares mean; SE, standard error; UACR, urinary albumin-to-creatinine ratio.

In contrast, A Study Comparing Dulaglutide with Insulin Glargine on Glycemic Control in Participants with Type 2 Diabetes and Moderate or Severe Chronic Kidney Disease (AWARD-7) was a clinical trial with dulaglutide in patients with T2D at high risk for kidney disease outcomes by virtue of entering the study with moderate-to-severe DKD (mean eGFR 38 mL/min/1.73 m$^2$) [55]. Participants (N = 577) were randomized (1:1:1) to receive insulin glargine, dulaglutide 0.75 mg once weekly or dulaglutide 1.5 mg once weekly. While the primary outcome of the trial was change in glycated hemoglobin A1c (A1C) at 26-weeks, secondary outcomes included change in eGFR and albuminuria over the 52-week total duration of the trial [55]. Both dulaglutide treatment groups experienced less eGFR decline compared to participants in the insulin glargine treatment group despite similar reductions in glycemia. Mean eGFR decline was −3.3 mL/min/1.73 m$^2$ in the insulin glargine group versus −0.7 mL/min/1.73 m$^2$ with both doses of dulaglutide [55]. A subsequent exploratory analysis of AWARD-7 found that participants treated with the 1.5 mg weekly dose of dulaglutide for 1 year had a significantly lower risk for ≥40% eGFR decline from baseline or progression to ESKD when compared to those treated with insulin glargine (HR: 0.45; 95% CI: 0.20–0.97; $p$ = 0.04) [58]. Notably, most events occurred in the subset of participants with severely increased albuminuria, where the risk of the composite outcome was greatly reduced with dulaglutide 1.5 mg versus insulin glargine (HR: 0.25; 95% CI: 0.10–0.68; $p$ = 0.006; Table 2) [58].

### 3.2. Exenatide

The Exenatide Study of Cardiovascular Event Lowering (EXSCEL) CVOT was a large trial (N = 14,752) that enrolled participants with T2D with (73%) or without (27%) established ASCVD [50]. The primary three-point MACE composite CV outcome was noninferior in participants randomized to once-weekly exenatide compared to the placebo cohort; however, statistical significance was not achieved to demonstrate superiority after a median follow-up of 3.2 years (Table 1) [50]. When examining kidney function change during the trial, eGFR change from baseline was similar in both the once-weekly exenatide and placebo treatment groups [59]. New-onset severely increased albuminuria (UACR > 300 mg/g) occurred in 2.2% of those randomized to exenatide and in 2.5% randomized to placebo (HR: 0.87; 95% CI: 0.70–1.1; $p$ = 0.19) [59]. No benefit on kidney composite outcomes were noted with exenatide treatment (Table 2).

Another post hoc analysis examined the impact of short-acting exenatide versus insulin glargine on markers of kidney disease in patients with T2D without a history of CKD [62]. Similar to the EXSCEL data described above, a beneficial effect of exenatide was not observed on either eGFR decline or progression of albuminuria [62]. In contrast, a smaller (N = 92) randomized trial enrolled participants with T2D with an eGFR ≥ 30 mL/min/1.73 m$^2$ and severely increased albuminuria (defined as 24 h urinary albumin excretion rate (UAER) > 0.3 g/24 h) [63]. Participants were randomized to receive exenatide twice-daily plus insulin glargine (intervention group) or insulin lispro plus insulin glargine (control group). The primary outcome was percent change in UAER from baseline to 24 weeks. At 24 weeks, the UAER decline was significantly larger in the exenatide twice-daily plus insulin glargine group when compared to the control group (−29.71%; 95% CI: −55.27 to −4.15%; $p$ = 0.0255) [63]. The authors posited that their findings may have differed from those reported by EXSCEL investigators due to differences in study populations, with this study enrolling participants with macroalbuminuria compared to EXSCEL which enrolled participants with less severe kidney disease at baseline.

### 3.3. Lixisenatide

The Evaluation of Lixisenatide in Acute Coronary Syndrome (ELIXA) trial enrolled participants (N = 6068) with T2D and a recent history of myocardial infarction or hospitalization for unstable angina [51]. Nearly 20% of participants in ELIXA had moderately increased albuminuria (UACR 30–300 mg/g), and approximately 6% had severely increased albuminuria (UACR > 300 mg/g). Lixisenatide achieved noninferiority for its primary

four-point MACE outcome when compared to placebo (HR: 1.0; 95% CI: 0.89–1.2; $p < 0.001$), but prespecified criteria to demonstrate superiority for MACE was not achieved (Table 1). An exploratory analysis examined changes in albuminuria and eGFR stratified by baseline albuminuria [60]. Additionally, time to new onset of macroalbuminuria and doubling of serum creatinine were assessed [60]. Lixisenatide treatment was associated with reduced progression of albuminuria in participants with severely increased albuminuria when adjusted for DKD risk factors (−39%; 95% CI −69 to −9.8; $p = 0.007$). A lower risk for new onset macroalbuminuria was observed with lixisenatide treatment when adjusted for baseline A1C (HR 0.81; 95% CI 0.66–0.99; $p = 0.04$; Table 2) [60]. No differences in eGFR decline were noted between the lixisenatide and placebo treatment arms [60].

### 3.4. Liraglutide

The Liraglutide Effect and Action in Diabetes: Evaluation of Cardiovascular Outcome Results (LEADER) trial (N = 9340) enrolled participants with T2D and high CV risk [52]. Approximately one-quarter of enrolled participants had moderate-to-severe DKD. The primary three-point MACE composite outcome occurred in 13% and 15% of participants randomized to liraglutide and placebo, respectively (Table 1) [40]. LEADER included a secondary kidney composite outcome that included new-onset severely increased albuminuria, doubling of serum creatinine, ESKD, or death due to kidney disease [61]. The composite kidney outcome occurred in 161 and 215 patients treated with liraglutide and placebo, respectively (HR: 0.78; 95% CI: 0.67 to 0.92; $p = 0.003$; Table 2). The decline in eGFR at 36 months was −7.4 versus −7.8 mL/min/1.73 m$^2$ in the liraglutide and placebo groups, respectively [61]. The LIRA-RENAL trial, with a shorter 26-week treatment duration, reported no benefit of liraglutide therapy on eGFR decline, possibly due to a shorter observation period [64].

### 3.5. Semaglutide

The Trial to Evaluate Cardiovascular and Other Long-term Outcomes with Semaglutide in Subjects with Type 2 Diabetes (SUSTAIN-6) enrolled patients (N = 3297) with established CV disease, CKD, or both [53]. Injectable semaglutide was superior to placebo for the primary three-point MACE composite outcome (Table 1). SUSTAIN-6 included a prespecified secondary kidney outcome of new or worsening "nephropathy," which was significantly reduced in the semaglutide group when compared to placebo (HR: 0.64; 95% CI: 0.46–0.88; $p = 0.005$; Table 2). The "nephropathy" benefit was largely driven by differences in albuminuria [53].

The Peptide Innovation for Early Diabetes Treatment (PIONEER)-6 trial assessed the CV safety of oral semaglutide in 3183 participants [54]. Oral semaglutide demonstrated noninferiority when compared to placebo for the primary three-point MACE composite outcome (HR: 0.79; 95% CI: 0.57–1.1; $p < 0.001$ for noninferiority). A larger CVOT with oral semaglutide adequately powered to test for CV superiority is currently underway, which includes prespecified secondary kidney outcomes [65].

## 4. Current Recommendations for GLP-1 Receptor Agonist Use in DKD

In consideration of current evidence for organ protective benefits with agents from the GLP-1 receptor agonist class, guidelines for the management of DM, CV disease, and kidney disease have evolved dramatically in recent years. Guideline-forming groups such as the American Diabetes Association (ADA), the American Association of Clinical Endocrinologists/American College of Endocrinology (AACE/ACE), the European Association for the Study of Diabetes (EASD), the European Society of Cardiology (ESC), and the Kidney Disease Improving Global Outcomes (KDIGO) all now recommend GLP-1 receptor agonists for glycemic control and ASCVD risk reduction in patients who have T2D with or without DKD [4,5,8,66–68].

Since GLP-1 receptor agonists are highly effective glucose-lowering agents in T2D, the ADA, EASD, ESC and KDIGO all recommend GLP-1 receptor agonists as an option for

patients to improve hyperglycemia and A1C [4,5,8,66–68]. Additional benefits of GLP-1 receptor agonists include a low risk of contributing to hypoglycemia and the potential for promoting weight loss. The ADA specifically recommends a GLP-1 receptor agonist in preference to insulin for patients with T2D who require greater glucose lowering than can be achieved with oral glucose-lowering agents [8]. KDIGO recommends first-line glucose-lowering therapy with metformin and a SGLT2 inhibitor in DKD [68]. For those requiring additional glucose-lowering to meet individualized glycemic targets, a GLP-1 receptor agonist is recommended as the preferred add-on glucose-lowering agent. This recommendation is based in part on emerging evidence for kidney benefits observed in large CVOTs (Table 2) and the preserved glucose-lowering effects of GLP-1 receptor agonists down to an eGFR of 15 mL/min/1.73 m$^2$ [68]. When selecting a glucose-lowering agent in the setting of T2D, a primary consideration per the 2022 ADA Standards of Medical Care in Diabetes is the presence of high-risk or established ASCVD, DKD, or HF [8]. If present, an agent with evidence for ASCVD, DKD and/or HF risk reduction is recommended for consideration independent of A1C or A1C target. Both ADA and AACE/ACE also advocate considering first-line use of glucose lowering agents with evidence of ASCVD, DKD or HF benefit in patients with these comorbidities (or indicators of high risk) at the time of T2D diagnosis [8,66]. Clearly, in patients who have a history or are considered high risk for ASCVD, the use of a GLP-1 receptor agonist or an SGLT2 inhibitor with proven CV benefit is preferred (Table 1) [8]. For albuminuric patients with T2D and CKD, the addition of an SGLT2 inhibitor is preferred per current ADA, EASD, ESC and KDIGO recommendations. If an SGLT2 inhibitor is contraindicated or clinically inappropriate, however, a GLP-1 receptor agonist is alternatively recommended. Per the ADA, in patients with T2D and CKD (eGFR < 60 mL/min/1.73 m$^2$) without albuminuria, either a GLP-1 receptor agonist or SGLT2 inhibitor with evidence of benefit can be utilized to mitigate CV risk [8].

## 5. Future Directions and Perspectives

GLP-1 receptor agonists are guideline-directed medical therapies for the management of T2D and ASCVD in patients with DKD. Evidence for use to protect the kidney in DKD are largely based on secondary outcomes from large CVOTs and the AWARD-7 clinical trial. The Effect of Semaglutide Versus Placebo on the Progression of Renal Impairment in Subjects with Type 2 Diabetes and Chronic Kidney Disease (FLOW) trial is a random-ized kidney disease outcome trial underway with injectable semaglutide [48]. FLOW-enrolled participants with T2D and kidney impairment are defined as either: (1) eGFR > 50 and ≤75 mL/min/1.73 m$^2$ with a UACR > 300 and <5000 mg/g; or (2) eGFR ≥ 25 and <50 mL/min/1.73 m$^2$ and UACR > 100 and <5000 mg/g. The primary study outcome is a composite kidney outcome inclusive of ≥50% eGFR decline, progression to ESKD or death from kidney or CV disease. FLOW is anticipated to be completed in 2024 and will foundationally inform the clinical role of GLP-1 receptor agonists in the management of DKD and future practice guidelines [56].

## 6. Conclusions

Even with the standard-of-care use of ACE inhibitors and ARBs, which are widely underutilized in the overall DKD population, the residual risk for the progression of kidney disease and CV events persists. Fortunately, recent advancements in the management of DKD provide clinicians and patients with novel therapeutic options for kidney and heart protection. While primary kidney outcome data are not yet available with agents from the GLP-1 receptor agonist class, secondary kidney outcome data are promising, with experimental evidence supporting the anti-inflammatory and anti-fibrotic properties of GLP-1 receptor agonists in the kidney. In the absence of primary kidney outcome data, GLP-1 receptor agonists are already recommended in patients with DKD to mitigate CV risk and improve glycemic control. Organizations such as the ADA even now recommend combination SGLT2 inhibitor plus GLP-1 receptor agonist therapy for additive organ

protection, based on patient-specific considerations [8]. Given their preserved glucose-lowering effects in the setting of DKD, combined with their demonstrated CV benefits, GLP-1 receptor agonists have a clear present role in the management of DKD. Pending kidney outcome trials will further define the role of GLP-1 receptor agonists in the management of DKD in the near future.

**Author Contributions:** Conceptualization, J.J.N.; bibliography research, J.J.N. and R.Z.A.; writing—original draft preparation, J.J.N.; writing—review and editing, J.J.N., R.Z.A. and K.R.T.; supervision, K.R.T. All authors have read and agreed to the published version of the manuscript.

**Funding:** This research received no external funding.

**Institutional Review Board Statement:** Not applicable.

**Informed Consent Statement:** Not applicable.

**Data Availability Statement:** Not applicable.

**Acknowledgments:** The authors would like to acknowledge Emily J. Cox for her expertise in illustration development.

**Conflicts of Interest:** J.J.N. reports personal fees and other support from Bayer AG; personal fees from Sanofi; personal fees from Novo Nordisk; and personal fees from Dexcom outside of the submitted work. R.Z.A. reports RZA reports grant support from Bayer AG, and Goldfinch Bio, and personal fees from Boehringer Ingelheim outside of the submitted work. K.R.T. reports other support from Eli Lilly; personal fees and other support from Boehringer Ingelheim; personal fees and other support from AstraZeneca; grants, personal fees and other support from Bayer AG; grants, personal fees and other support from Novo Nordisk; grants and other support from Goldfinch Bio; other support from Gilead; and grants from Travere outside the submitted work.

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
