# Peer review of "GLP-1 Receptor Agonists in the Treatment of Patients with Type 2 Diabetes and Chronic Kidney Disease"

_kidneydial, doi:10.3390/kidneydial2030034_

Round 1

Reviewer 1 Report

The authors showed the role of GLP-1 receptor agonists in DKD.

Should the author add a detailed mechanism of the role of GLP-1 receptor agonists in DKD.

Author Response

  1. The authors showed the role of GLP-1 receptor agonists in DKD.

Thank you.

  1. Should the author add a detailed mechanism of the role of GLP-1 receptor agonists in DKD.

Thank you. We have addressed proposed mechanisms of kidney benefit in section 2 of the manuscript. Please see edits/additions to this section as recommended. We have also included a review of current recommendations for GLP-1 receptor agonist use in the setting of DKD from a clinical perspective.

Reviewer 2 Report

The paper of Joshua J. Neumiller and coworkers is an interesting update of a relevant topic, i.e. the therapeutic use of GLP-1 receptor agonists in diabetic kidney disease. The review is well written and represents a comprehensive discussion of the mechanisms of renoprotection of GLP-1 receptor agonists and findings from large prospective outcome trials.

However, the authors have missed some points and should mention them to make the review fully exhaustive:

Part 2. Proposed Mechanisms for Kidney Benefit with GLP-1 Receptor Agonists

The inhibition of oxidative stress, inflammation, fibrosis, and induction of natriuresis have been implicated as mechanisms underlying the attenuation of DKD by GLP-1 receptor agonists. However, in the paper, the author only discussed the inhibition of inflammation and oxidation. It would be more comprehensive to comment on other mechanisms like fibrosis and induction of natriuresis. This would deserve one paragraph.

Also, I doubt if reprinted images are works as it may affect the novelty of the review. It would be better for the author to create new images using some tools like Biorender.

Author Response

  1. The paper of Joshua J. Neumiller and coworkers is an interesting update of a relevant topic, i.e., the therapeutic use of GLP-1 receptor agonists in diabetic kidney disease. The review is well written and represents a comprehensive discussion of the mechanisms of renoprotection of GLP-1 receptor agonists and findings from large prospective outcome trials. However, the authors have missed some points and should mention them to make the review fully exhaustive.

Thank you.

  1. Part 2. Proposed Mechanisms for Kidney Benefit with GLP-1 Receptor Agonists. The inhibition of oxidative stress, inflammation, fibrosis, and induction of natriuresis have been implicated as mechanisms underlying the attenuation of DKD by GLP-1 receptor agonists. However, in the paper, the author only discussed the inhibition of inflammation and oxidation. It would be more comprehensive to comment on other mechanisms like fibrosis and induction of natriuresis. This would deserve one paragraph.

We have more specifically called out effects on fibrosis and added additional discussion on diuresis/natriuresis as a proposed mechanism of GLP-1 receptor agonist benefit. These proposed mechanisms are also highlighted within Figures 1 and 2.

  1. Also, I doubt if reprinted images are works as it may affect the novelty of the review. It would be better for the author to create new images using some tools like Biorender.

The submitted images were recently generated by our team and align with the content of this review. We believe these figures are informative and add to the submitted review and it would be difficult to improve/expand upon them for the purposes of this review. If the editorial team would prefer not to run reprinted figures, however, we are happy to remove the figures and update the text accordingly. Thank you.

Round 2

Reviewer 1 Report

My previous comments have been addressed.